# RFTF: REINFORCEMENT FINE-TUNING FOR VISION-LANGUAGE-ACTION MODELS WITH TEMPORAL FEEDBACK

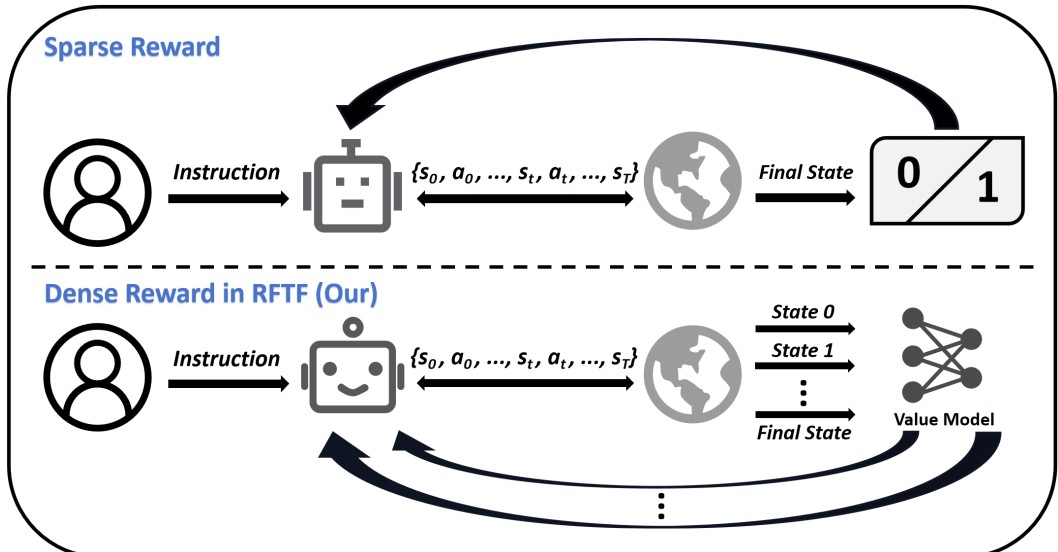

Figure 1: **Comparison between sparse reward and dense reward.** In typical reinforcement fine-tuning methods for VLAs, only sparse, outcome-based rewards are provided, which can confuse VLAs when encountering partially correct or incorrect episodes. In contrast, RFTF leverages a value model trained with temporal information to predict the value of each state within an episode, providing VLAs with higher-granularity dense rewards.

## ABSTRACT

Vision-Language-Action (VLA) models have demonstrated significant potential in the field of embodied intelligence, enabling models to follow human instructions to complete complex tasks in physical environments. Existing VLAs are often trained through behavior cloning, which requires expensive data and computational resources and is constrained by human demonstrations. To address this issue, many researchers explore the application of reinforcement fine-tuning to VLAs. However, typical reinforcement fine-tuning methods for VLAs usually rely on sparse, outcome-based rewards, which struggle to provide fine-grained feedback for specific actions within an episode, thus limiting the model's manipulation capabilities and generalization performance. In this paper, we propose RFTF, a novel reinforcement fine-tuning method that leverages a value model to generate dense rewards in embodied scenarios. Specifically, our value model is trained using temporal information, eliminating the need for costly robot action labels. In addition, RFTF incorporates a range of techniques, such as GAE and sample balance to enhance the effectiveness of the fine-tuning process. By addressing the sparse reward problem in reinforcement fine-tuning, our method significantly improves the performance of VLAs, delivering superior generalization and adaptation capabilities across diverse embodied tasks. Experimental results

show that VLAs fine-tuned with RFTF achieve new state-of-the-art performance on the challenging CALVIN ABC-D with an average success length of 4.296. Moreover, RFTF enables rapid adaptation to new environments. After fine-tuning in the D environment of CALVIN for a few episodes, RFTF achieved an average success length of 4.301 in this new environment.

# 1 INTRODUCTION

The field of embodied intelligence has made remarkable progress in recent years [25]. Vision-Language-Action models, by integrating visual perception, language understanding, and action execution, empower models to perform a variety of tasks in the physical world following human instructions [27]. However, training VLAs directly through behavior cloning requires vast and costly labeled data and computational resources [6; 33], and is limited by the scope of human demonstrations. As a result, VLAs often exhibit suboptimal manipulation and generalization capabilities in practical applications.

To enhance the generalization and reasoning abilities of VLAs while reducing reliance on extensive data and computational resources, increasing research efforts have focused on reinforcement fine-tuning for VLAs [9; 35; 14; 12]. Reinforcement learning (RL) trains models through trial-and-error, allowing them to learn from past experiences without depending on human-provided data. In addition, RL allows models to adapt to new environments by trying out and updating the parameters without human intervention. However, a significant challenge in reinforcement fine-tuning for VLAs is the reliance on sparse, outcome-based rewards. Such reward mechanisms fail to capture the nuanced correctness of individual actions within an episode. For instance, when there is partial correctness or incorrectness, it can lead to erroneous encouragement or suppression of certain actions, ultimately compromising the efficiency and stability of reinforcement fine-tuning [26].

In this paper, we propose RFTF, a reinforcement fine-tuning method with dense-reward from temporal feedback for VLAs. RFTF contains two stages. Specifically, in the first stage, we present a value model tailored for embodied scenarios. This value model is trained using temporal information to predict the value of the current state based on the input state and human instruction. In the second stage, we integrate this value model into an RL fine-tuning framework for VLAs based on Proximal Policy Optimization (PPO) [37]. By combining reward shaping and generalized advantage estimation (GAE) [36], we ensure an efficient reinforcement fine-tuning process. Notably, the entire training process for the value model and VLAs does not require robot action labels, showing the proposed method's potential for efficient data utilization.

The contributions of this paper are as follows:

- We introduce a dense-reward reinforcement fine-tuning method for VLAs without any robot action labels from humans.
- We present a value model trained with temporal information to generate dense rewards and combine reward shaping and GAE strategy to facilitate the RL fine-tuning process.
- Experimental results on the CALVIN benchmark [30] show that the proposed method achieves new state-of-the-art performance under the ABC-D setting. Moreover, RFTF exhibits superior adaptation capabilities in a new environment.

# 2 RELATED WORK

## 2.1 FOUNDATION MODELS FOR EMBODIED INTELLIGENCE

In recent years, large language models (LLMs) [1; 46; 42; 11] and vision-language models (VLMs) [18; 23; 10; 40; 2] trained on web-scale data have not only demonstrated the ability to engage in human-like dialogue but also exhibited remarkable reasoning capabilities and understanding of the physical world [16; 39; 44; 22; 29]. Leveraging these strengths, many researchers explore how to apply foundation models to interact with the physical world, *i.e.*, for embodied intelligence.

Broadly, there are two approaches to achieve physical environment interaction with foundation models: high-level planning and low-level manipulation. For high-level planning, LLMs or VLMs are

used for planning to capitalize on their robust comprehension and reasoning abilities to translate complex human instructions into simpler robot skills [5; 15; 17; 28; 45], *e.g.*, move to, grasp, and so on. However, directly utilizing LLMs or VLMs does not yield control signals for embodiments and requires additional models to complete simpler robot skills. For low-level manipulation, the output of the pretrained VLMs is changed to action and construct VLAs, enabling direct interaction with the physical environment. Specifically, RT-2 [48] fine-tunes PaLI-X [8] on both vision-language data and robot demonstrations to build the robot policy. OpenVLA [20] fine-tunes Prismatic on Open X-Embodiment (OXE) [34] dataset, which contains 22 different robotic embodiments from 21 different institutions. $\pi_0$ [4] proposes a novel flow matching architecture built on top of the PaliGemma VLM [3] to inherit Internet-scale semantic knowledge. It is worth noting that not all VLAs are derived from pre-trained VLMs. RDT [24] builds on diffusion models to perform complex bimanual manipulation. Seer [41] generates actions through an inverse dynamics model that is conditioned on the robot's anticipated visual states.

However, these models are trained with human action labels in a behavior cloning way. In this paper, we train VLAs through reinforcement fine-tuning to improve their generalization and help them adapt to novel environments with any action labels.

## 2.2 REINFORCEMENT FINE-TUNING FOR LARGE MODELS

Reinforcement learning is a technique that enables learning from a model's past experiences. Unlike supervised learning, RL does not require large amounts of manually annotated data, nor is it constrained by expert demonstrations. Recently, using RL to fine-tune pretrained large language models has become a trend, enhancing models' reasoning capabilities or aligning them with human preferences [32; 11; 38; 43; 47]. However, reinforcement fine-tuning in the field of embodied intelligence differs from that in LLMs, as it necessitates extensive interaction with the environment to collect data. FLaRe [14] is a large scale reinforcement fine-tuning framework that introduces a series of design choices that help stabilize the RL training process. iRe-VLA [12] iterates between reinforcement learning and supervised learning to address the instability issues of reinforcement learning in large-scale VLAs. DPPO [35] improves diffusion-based policies by leveraging the sequential nature of the diffusion denoising process and fine-tuning the entire chain of diffusion MDPs.

Nonetheless, due to the high precision requirements for output actions, directly applying sparse rewards in the reinforcement fine-tuning of VLA models often yields suboptimal results and may even lead to performance degradation. In contrast, we incorporate dense rewards into the reinforcement fine-tuning of VLAs through a value model trained with temporal information.

## 3 METHOD

Our objective is to utilize a value model to provide dense rewards for reinforcement fine-tuning of VLAs, thereby enhancing their generalization and adaptation capabilities. In Notation and preliminary Section3.1, we begin by introducing the notation and preliminary used in our approach. In Value model Section 3.2, we introduce the structure of the value model and the methodology for its training. In RL fine-tuning pipeline Section3.3, we describe how the value model is applied to the reinforcement fine-tuning process for VLAs.

### 3.1 NOTATION AND PRELIMINARY

Due to challenges such as limited camera coverage and occlusions between objects, we model each robot task as a Partially Observable Markov Decision Process (POMDP), defined by the tuple $(\mathcal{S}, \mathcal{A}, \mathcal{P}, \mathcal{R}, \mathcal{O}, \mathcal{L}, \gamma)$. Here, $\mathcal{S}$ and $\mathcal{A}$ represent the state space and action space, respectively. $\mathcal{P} : \mathcal{S} \times \mathcal{A} \times \mathcal{S} \to [0, 1]$ denotes the state transition probability function. $\mathcal{R} : \mathcal{S} \times \mathcal{A} \times \mathcal{L} \times \mathcal{S} \to \mathbb{R}$ is the reward function. $\mathcal{O}$ is the observation space, specifically consisting of RGB images from different cameras. $\mathcal{L}$ is the set of human instructions guiding the robot to complete tasks. $\gamma \in [0, 1]$ is the discount factor, controlling the balance between short-term and long-term focus.

We denote the current robot policy as $\pi_\theta$, *i.e.*, a Vision-Language-Action model parameterized by $\theta$. The policy $\pi_\theta$ takes the state (*e.g.*, observation and robot state) as input and outputs the corre-

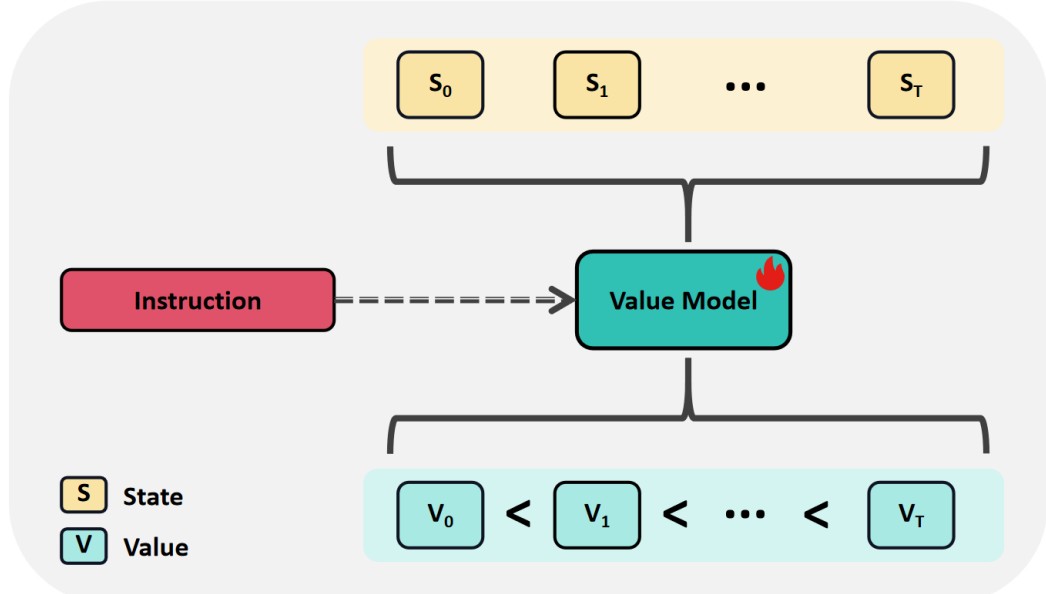

Figure 2: **Training procedure of the value model**. We assume that during an episode of a human-demonstrated successful embodied task, the state value increases monotonically over time.

sponding action. The overall optimization objective is to maximize the expected return of $\pi_\theta$ with the discount factor $\gamma$, *i.e.*, $J(\theta) = \mathbb{E}_{(s_t,a_t)\sim P} \sum_t \gamma^t R(s_t, a_t)$.

## 3.2 VALUE MODEL

Vision-language-action models take the human instructions and state at each time step as input, outputting low-level robot actions to interact with the environment and complete tasks. An embodied episode may consist of hundreds or thousands of actions. Current reinforcement fine-tuning methods applied to VLAs, such as FLaRe [14] and DPPO [35], predominantly rely on sparse, outcome-based rewards, which are inadequate when the decisions of VLAs exhibit local correctness or errors.

To automatically label the correctness of each action in an episode and provide dense reward for training, we employ a value model parameterized by $\phi$ to predict the value of the state at each time step. Specifically, the value model takes the state and human instruction as input and outputs the value of the current state, *i.e.*, $v_t = V_\phi(s_t, l)$. Notably, both the inference and training of the value model do not require expensive robot action labels, resulting in low data dependency.

However, there are no explicit value labels available for training the value model. Inspired by RLHF [32], we collect data pairs and train the value model with contrastive learning. Specifically, as shown in Fig. 2, $(s_t, s_{t+1}, ..., s_{t+n-1} \mid l)$ is an expert-demonstrated trajectory without action labels. As the state progresses toward task completion, we assume that the state value increases with each time step, *i.e.*, $v_t < v_{t+1} < ... < v_{t+n-1}$. Following RLHF, we adopt the contrastive loss function as the optimization objective:

$$loss(\phi) = -\frac{1}{C_n^2}\mathbb{E}_{(s_t,a_t)\sim P}[log(\sigma(V_\phi(s_{t+\Delta t}, l) - V_\phi(s_t, l)))], \tag{1}$$

where $C_n^2$ is the number of combinations, $\sigma$ is the sigmoid function and $\Delta t$ is a positive integer belonging to $[1, n-t)$. This implies that we aim for the value at later time steps to be as large as possible compared to earlier time steps.

The architecture of our value model is based on the VLA model, with only the action tokens of the VLA model replaced by value tokens. Given that the VLA model is already capable of processing

mixed inputs of states and human instructions, we initialize the training of the value model using the weights of the VLA model.

## 3.3 RL FINE-TUNING PIPELINE

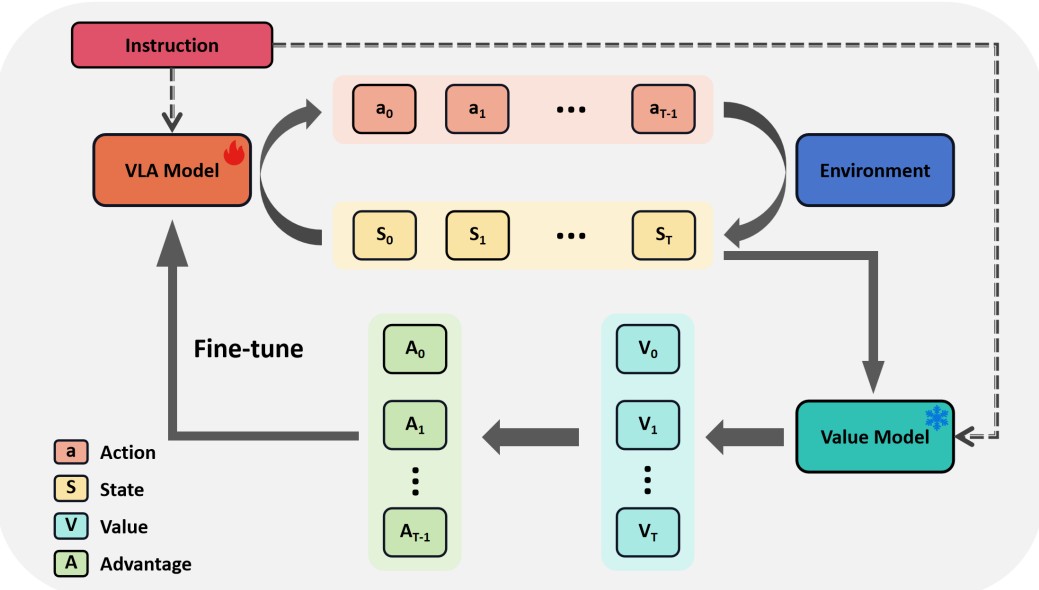

Figure 3: **Illustration of RL fine-tuning pipeline.** RFTF utilizes a value model trained with temporal information to predict the value of each state in episodes of interaction between the VLA model and the environment, thereby providing guidance for each action in episodes to fine-tune the VLA model.

In this stage, as shown in Fig. 3, we utilize the trained value model to guide the RL fine-tuning process, aiming to enhance the performance of VLAs. To achieve effective fine-tuning, we adopt Proximal Policy Optimization (PPO) [37] as our reinforcement learning algorithm framework.

Specifically, after obtaining the raw value of each state in an episode from the trained value model, we normalize all state values within the episode, as the output range of the value model varies significantly across different tasks. To make progress toward the final goal, we employ a reward shaping term [31] as the reward function:

$$R_t = \begin{cases} \gamma V(s_{t+1}, l) - V(s_t, l) & \text{if } t \text{ is not the end step} \\ 0 & \text{otherwise} \end{cases} \tag{2}$$

Notably, if we directly use the discounted sum of rewards with the discount factor $\gamma$ as the advantage function in PPO, the simplified form of the advantage function will contain only the current and last states, thereby neglecting the influence of intermediate states. To address this, we adopt Generalized Advantage Estimation (GAE) [36], introducing an additional hyperparameter $\lambda$ to incorporate the state values at all intermediate time steps into the advantage function. Additionally, to leverage information about task completion, instead of solely adding +1 or -1 to the reward at the final time step, we incorporate feedback on task success or failure into the advantage function at each time step. This ensures that early decisions in long episodes receive relevant feedback. Furthermore, we introduce a balancing coefficient to address the imbalance between successful and failed samples. The final form of our advantage function is as follows:

$$A_t = \eta \left[ I(success) + \sum_{n=t}^{T} (\gamma\lambda)^{n-t} R_t \right], \tag{3}$$

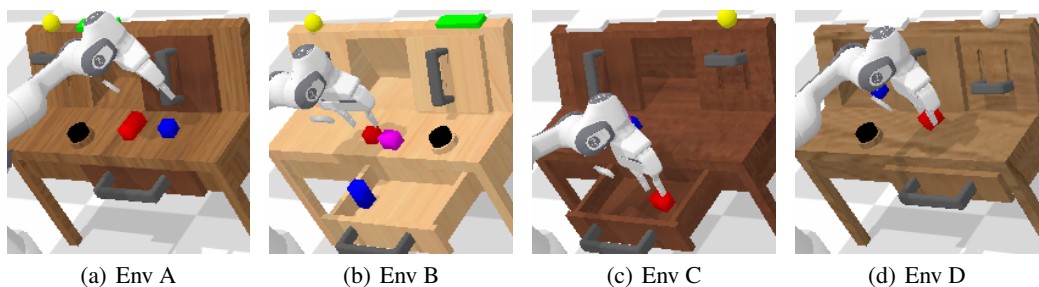

(a) Env A         (b) Env B         (c) Env C         (d) Env D

Figure 4: **Visualization of the CALVIN benchmark**. The CALVIN benchmark includes four distinct environments, differing in the positions of the LED, light bulb, slider, drawer, switch, and button, as well as the material of the table.

where $\eta$ is the coefficient for balancing positive and negative samples, set to 0.25 when the task succeeds and 1 when the task fails. $\lambda$ is the hyperparameter in GAE. $I$ is an indicator function that returns +1 for task success and -1 for task failure.

Previous works [14; 12; 9] have highlighted that, due to the high precision requirements for VLA model outputs, directly applying reinforcement fine-tuning to VLAs often results in performance degradation. To mitigate this, inspired by [38], we incorporate both a surrogate objective clipping term and adaptive KL divergence into the optimization objective of reinforcement fine-tuning. Specifically, for an VLA model $\pi$ parameterized by $\theta$, the loss function for reinforcement fine-tuning is as follows:

$$
\begin{aligned}
loss(\theta) = -\mathbb{E}_{(s_t, a_t) \sim P} \Big\{ & min \Big[ \frac{\pi_\theta(a_t|s_t)}{\pi_{\theta_{old}}(a_t|s_t)} A_t, \\
& clip \Big( \frac{\pi_\theta(a_t|s_t)}{\pi_{\theta_{old}}(a_t|s_t)}, 1 - \epsilon, 1 + \epsilon \Big) A_t \Big] \\
& -\beta D_{KL} \big[ \pi_\theta(a_t|s_t) || \pi_{\theta_{ref}}(a_t|s_t) \big] \Big\},
\end{aligned}
\tag{4}
$$

where $\epsilon$ and $\beta$ are hyper-parameters that prevent excessive divergence between the new and old policies.

## 4 EXPERIMENTS

### 4.1 EXPERIMENTAL SETUP

#### 4.1.1 BENCHMARK

We conducted experiments on the challenging CALVIN benchmark. CALVIN focuses on long-horizon robotic tasks conditioned on language, encompassing 34 tasks across four distinct simulated environments (Env A, B, C, and D). As shown in Fig. 4, each environment features a Franka Emika Panda robot equipped with a parallel-jaw gripper and a table for performing various manipulation tasks. Leveraging the four diverse environments of the CALVIN benchmark, we can conduct both generalization and adaptation experiments to validate the effectiveness of our approach.

#### 4.1.2 BASELINES

In RFTF, we fine-tune the top two models on the CALVIN ABC-D benchmark under two distinct settings. In the generalization setting, we fine-tune the models on the CALVIN ABC environments, which the models have already encountered, to evaluate whether RFTF enhances the models' generalization performance. In the adaptation setting, we fine-tune the models on the CALVIN D environment, which the models have not previously seen, to assess whether RFTF helps the models adapt to new environments.

Table 1: **CALVIN ABC-D results.** We present the average success rates of top-3 checkpoints computed over 1000 rollouts for each task and the average number of completed tasks to solve 5 instructions consecutively (Avg. Len.). The model being fine-tuned is specified in parentheses after RFTF.

| Method | Task completed in a row | | | | | |
|---|---|---|---|---|---|---|
| | L1 | L2 | L3 | L4 | L5 | Avg. Len. ↑ |
| 3D Diffuser Actor [19] | 92.2 | 78.7 | 63.9 | 51.2 | 41.2 | 3.272 |
| CLOVER [7] | 96.0 | 83.5 | 70.8 | 57.5 | 45.4 | 3.532 |
| Diffusion Transformer Policy [13] | 94.5 | 82.5 | 72.8 | 61.3 | 50.0 | 3.611 |
| Seer [41] | 94.4 | 87.2 | 79.9 | 72.2 | 64.3 | 3.980 |
| GR-MG [21] | 96.8 | 89.3 | 81.5 | 72.7 | 64.4 | 4.047 |
| **RFTF(GR-MG)** | **96.9** | **88.8** | **82.1** | **74.9** | **65.4** | **4.081** |
| Seer-Large [41] | 96.3 | 91.6 | 86.1 | 80.3 | 74.0 | 4.283 |
| **RFTF(Seer-Large)** | **96.4** | **91.7** | **86.7** | **80.7** | **74.1** | **4.296** |

### 4.1.3 METRICS

The CALVIN benchmark evaluation requires VLAs to execute 1000 sequences in the D simulated environment, with each sequence comprising 5 tasks. The VLA model performs these 5 tasks sequentially, exiting the sequence upon failing any task. Ln denotes the proportion of n tasks completed out of 5. We use the average number of tasks completed per sequence as the evaluation metric.

### 4.1.4 IMPLEMENTATION DETAILS

For the value model, we train it with a batch size of $4 \times 8$ and a learning rate of 1e-5. As described in Value model Section3.2, the structure of the value model involves replacing the original VLA's action tokens and action decoder with value tokens and a value decoder.

For the RL fine-tuning, we apply the same implementation details across both the generalization and adaptation settings to ensure the method's universality. For the VLA model, we first discretize the model's output with 1000 bins to obtain the probability term in the PPO optimization objective. To enhance training stability, we freeze the model's encoders and transformer backbone, and only update the action head. During RL fine-tuning, we train the model with a learning rate of 1e-7, covering roughly 1000 episodes. The RL fine-tuning process is done with four NVIDIA A40 GPUs within 10 hours for Seer-Large and 14 hours for GR-MG. To prevent overfitting to task instructions, we deliberately used different instructions during the fine-tuning phase than those used during the testing phase.

### 4.2 MAIN RESULTS

We validate the effectiveness of our algorithm from two perspectives: first, RFTF enhances the generalization capabilities of VLAs; second, RFTF facilitates adaptation to new environments. Notably, the baseline models used for fine-tuning were trained solely on demonstration data from CALVIN's A, B, and C environments, without any data from the D environment. To ensure the reliability of the results, each experiment was evaluated using three different seeds, with the mean value reported as the final result.

### 4.2.1 GENERALIZATION

In this experiment, we show the better generalization ability of VLAs fine-tuned with RFTF. Specifically, RFTF fine-tunes models on CALVIN's A, B, and C environments and tests them in the D environment. As shown in Tab. 1, GR-MG fine-tuned by RFTF achieved a score of 4.081, surpassing the baseline of 4.043; Seer-Large fine-tuned by RFTF achieved a score of 4.296, surpassing the baseline of 4.283, which also achieves new state-of-the-art performance.

### 4.2.2 ADAPTATION

Table 2: **Adaptation Experiments.** VLA refers to models to be fine-tuned. Env denotes the environments we use to fine-tune the model, where "–" indicates no fine-tuning.

| VLA | Env | Task completed in a row | | | | | |
|-----|-----|------|------|------|------|------|-------------|
| | | L1 | L2 | L3 | L4 | L5 | Avg. Len. ↑ |
| GR-MG | - | 96.8 | 89.3 | 81.5 | 72.7 | 64.4 | 4.047 |
| GR-MG | ABC | 96.9 | 88.8 | 82.1 | 74.9 | 65.4 | 4.081 |
| **GR-MG** | **D** | **96.1** | **90.5** | **83.9** | **75.0** | **65.8** | **4.113** |
| Seer-Large | - | 96.3 | 91.6 | 86.1 | 80.3 | 74.0 | 4.283 |
| Seer-Large | ABC | 96.4 | 91.7 | 86.7 | 80.7 | 74.1 | 4.296 |
| **Seer-Large** | **D** | **97.0** | **92.0** | **86.0** | **80.6** | **74.5** | **4.301** |

In this experiment, we demonstrate that VLAs can be adapted to new environments that they have never seen before with the proposed RFTF. Specifically, we fine-tune models with RFTF in the unseen D environment of CALVIN. As indicated in Tab. 2, GR-MG fine-tuned by RFTF achieved a score of 4.113, and Seer-Large fine-tuned by RFTF achieved a score of 4.301, significantly outperforming the model's original performance in CALVIN's D environment. Since other methods are exclusively trained on the A, B, and C environments in CALVIN and have no exposure to the D environment, we refrain from comparing them with the results in our adaptation setting.

### 4.3 ANALYSIS OF VALUE MODEL

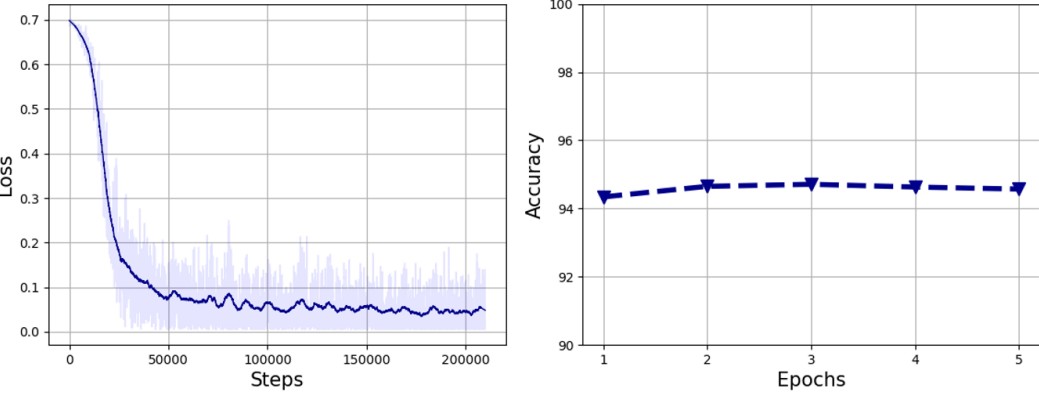

Figure 5: **Train curves of the value model.** We show the loss curve of the value model during training and evaluation results of the value model across different epochs.

To verify whether the value model can accurately predict state values, we tested its accuracy on the CALVIN ABC validation set. Specifically, we selected two frames from expert-demonstrated trajectories, and if the value model assigned a higher value to the later frame compared to the earlier one, we considered the value prediction correct.

As shown in Fig. 5, the value model achieved an accuracy of more than 94% after the first epoch, with subsequent training nearly yielding no further improvements in accuracy. This observation aligns with findings in RLHF. To mitigate potential overfitting from prolonged training, we selected the value model from the first epoch for use in the reinforcement fine-tuning process.

Notably, as shown in Fig. 6, we found that in episodes sampled by the VLA model itself, the state values produced by the value model did not exhibit a monotonic increase over time. This ensures that the optimization objective of our reinforcement fine-tuning does not merely encourage the VLA model to reinforce its original actions.

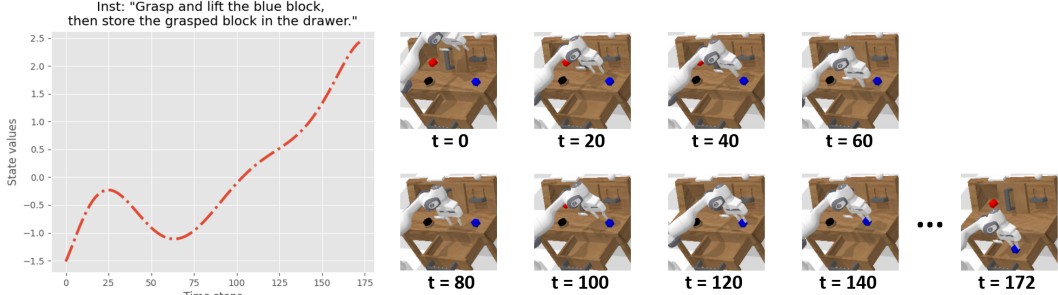

Figure 6: **An example of a state value curve.** As depicted, the curve exhibits a decline midway due to an incorrect grasping action by the VLA model.

Table 3: **Ablation studies.** VLA refers to models to be fine-tuned. SR refers to sparse reward. Replacing the dense rewards provided by RFTF with sparse rewards leads to performance drops.

| VLA | Type | Reward | Task completed in a row | | | | | |
|---|---|---|---|---|---|---|---|---|
| | | | 1 | 2 | 3 | 4 | 5 | Avg. Len. ↑ |
| GR-MG | Generalization | **RFTF** | **96.9** | **88.8** | **82.1** | **74.9** | **65.4** | **4.081** |
| GR-MG | Generalization | SR | 95.3 | 88.3 | 80.8 | 72.1 | 62.8 | 3.993 |
| GR-MG | Adaptation | **RFTF** | **96.1** | **90.5** | **83.9** | **75.0** | **65.8** | **4.113** |
| GR-MG | Adaptation | SR | 95.9 | 88.3 | 79.8 | 72.4 | 64.5 | 4.009 |
| Seer-Large | Generalization | **RFTF** | **96.4** | **91.7** | **86.7** | **80.7** | **74.1** | **4.296** |
| Seer-Large | Generalization | SR | 95.2 | 89.9 | 85.1 | 79.4 | 72.9 | 4.225 |
| Seer-Large | Adaptation | **RFTF** | **97.0** | **92.0** | **86.0** | **80.6** | **74.5** | **4.301** |
| Seer-Large | Adaptation | SR | 95.3 | 90.7 | 85.8 | 79.8 | 73.3 | 4.249 |

## 4.4 ABLATION STUDY

We conducted ablation experiments to evaluate the effectiveness of the dense rewards in RFTF. To ensure a fair and controlled comparison, all experimental conditions were kept identical except for the rewards used for fine-tuning. The sparse rewards for fine-tuning are derived solely from whether the model successfully completed the given task or not, which is a common method used in standard PPO algorithms.

As shown in Tab. 3, unlike models fine-tuned with RFTF, the models using only sparse rewards exhibited varying degrees of performance drop. This finding is consistent with the observations in [14] and [12].

## 5 CONCLUSION AND LIMITATION

In this paper, we propose RFTF, an online reinforcement fine-tuning method for vision-language-action models. To obtain dense rewards, we first train a value model using temporal information while maintaining low data dependency. Then, we integrate the value model into the reinforcement fine-tuning process for VLAs, providing reward signals for intermediate decision steps, addressing the prevalent issue of sparse rewards, and enhancing the effectiveness of fine-tuning. Experimental results demonstrate that VLAs fine-tuned with RFTF exhibit superior generalization and overall performance. Additionally, RFTF enables rapid adaptation to new environments. The primary limitation of RFTF is that it has only been verified on the simulated benchmark. In the future, we will apply RFTF to real-world robots.

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
