# OpenReview forum: "RFTF: Reinforcement Fine-tuning for Vision-language-action Models with Temporal Feedback"
_ICLR.cc/2026/Conference — ICLR 2026 Conference Withdrawn Submission_

### Official Review · Reviewer_XHsj · 2025-10-24

**Soundness:** 2
**Presentation:** 2
**Contribution:** 2
**Rating:** 2
**Confidence:** 4

**Summary:**

This paper proposes RFTF, a reinforcement fine-tuning pipeline for Vision–Language–Action (VLA) models that replaces sparse outcome rewards with dense, temporally informed rewards produced by a learned value model. The value model is trained from demonstration trajectories using a pairwise/contrastive objective that enforces vt < vt+∆t for later frames (Eq.1). During fine-tuning the value model’s per-step outputs are normalized and converted into shaped rewards Rt = γV(st+1) − V(st) (Eq.2); advantages use GAE with a success/failure bias term (Eq.3) and PPO with clipping + adaptive KL (Eq.4). Experiments are on the CALVIN ABC–D benchmark: RFTF applied to GR-MG and Seer-Large yields small improvements in average completed tasks (e.g., Seer-Large 4.283 → 4.296) and is reported to improve adaptation when fine-tuned in unseen environment D.

**Strengths:**

•	This paper proposes an intuitive and feasible remedy to alleviate the practical limitation of current VLA fine-tuning (sparse result rewards).

•	Provides concrete algorithmic components (contrastive value loss, reward shaping, PPO integration with GAE) and implementation details (discretization, learning rates, GPUs/time). These make the work actionable for other practitioners.

**Weaknesses:**

•	Incremental novelty: core ideas (temporal ranking → reward shaping) are not new; the paper mostly packages them for VLAs without clear theoretical or algorithmic advances beyond engineering. Cite earlier work on visual reward learning / temporal ranking as context and contrast more directly.

•	Method clarity: crucial design choices (pairwise sampling strategy, normalization, hyperparameter values and their selection) are insufficiently formalized; the writeup lacks compact pseudocode tying all pieces together.

•	Experiments conducted solely on the simple CALVIN benchmark are insufficient to demonstrate the effectiveness and generalizability of our approach. Specifically, we need to determine whether the proposed method is also effective for more challenging tasks, such as insertion and rotation, which are not part of the grasping and placement task. Furthermore, experiments based solely on the GR-MG and Seer models are insufficient; more advanced models, such as OpenVLA-OFT and pi0, should be considered.

•	Ablations and sensitivity checks are limited: missing (1) correlation analyses between predicted value and episode return, (2) sensitivity to discretization bins and normalization, (3) ablation of freezing backbone vs fine-tuning more of the model, (4) more robust baselines such as learned contrastive reward models or alternative dense-reward constructions.

•	Reproducibility & scope: while training cost reported (e.g., 4 A40 GPUs for 10–14 hours) is reasonable, many implementation details are in prose. Also, experiments are limited to simulation (authors acknowledge this) so real-robot applicability remains untested.

**Questions:**

1.	Provide a quantitative correlation analysis between per-step predicted values and episode returns / final success (scatterplots, correlation coefficients). This would validate that the learned value signal meaningfully predicts downstream reward.

2.	Ablate the major heuristic choices: discretization bin count, freezing backbone vs unfreezing varying fractions, the η balancing coefficient values, how ∆t is sampled for pairwise loss, and the effect of normalizing value outputs per episode. Show sensitivity curves.

3.	Compare RFTF to stronger learned-reward baselines (e.g., contrastive reward regressors, inverse dynamics / distance-to-goal estimates, or off-policy learned shaping rewards). If these baselines match or beat RFTF, that would weaken the novelty claim.

4.	Provide representative failure cases where the value signal is misleading (non-monotonic demonstrations, distractor motions) and describe mitigation strategies. The paper notes non-monotonicity in policy-sampled episodes (Fig.6) but does not quantify how often this occurs or its downstream effect.

---

### Official Review · Reviewer_1GME · 2025-10-26

**Soundness:** 2
**Presentation:** 1
**Contribution:** 2
**Rating:** 2
**Confidence:** 5

**Summary:**

This paper proposes RFTF, a two-stage recipe for densifying rewards when RL fine-tuning VLA policies on CALVIN. Stage 1 learns an instruction-conditioned value/potential $V_\phi(s,l)$ from within-demo temporal ordering, i.e. by ranking later demonstration frames above earlier ones (no action labels; value model initialized from the VLA). Stage 2 uses classic potential shaping $R_t=\gamma V_\phi(s_{t+1},l)-V_\phi(s_t,l)$ inside PPO, with two heuristics: per-episode normalization of the potential and a modified GAE over the potential-shaped rewards that adds a success indicator and a class-imbalance weight $\eta$ (0.25 on success, 1 on failure). Empirically, the method produces very modest but positive gains on CALVIN for two base VLAs.

**Strengths:**

Simple, implementable recipe.

General reward densification (uses temporal order), no task-specific hacks.

Modest gains on CALVIN.

**Weaknesses:**

RFTF packages well-known components (temporal ranking, learning potentials from demonstrations, potential shaping) into a VLA fine-tuning recipe, to solve the problem of sparse rewards. This could be a valuable contribution if the prevailing evidence in the literature lacked indication that these ideas could be successfully applied to VLAs, and if the existing approaches to training VLAs lacked a tested solution to the problem of sparse rewards or it provided strong, statistically significant empirical gains over existing solutions. However, the manuscript: (A) omits and fails to compare against directly relevant prior work (notably VLAC, arXiv:2509.15937), which packages a very similar recipe, (B) rests on a brittle monotonicity prior, the scope of which is neither justified nor stress-tested, and (C) presents very modest, statistically flimsy empirical gains.

The core issues are discussed below:

- Related-work omissions materially distort novelty. The method’s core prior (“later > earlier” -> dense progress signal) already appears in prior works. Not citing or comparing to such methods overstates contribution and deprives readers of a calibrated baseline. The authors do not discuss or address the plethora of adjacent approaches for VLA reward densification (e.g. VLM-RM, RL-VLM-F, RoboCLIP, ...), giving a misleading impression of novelty in tackling this problem.

- Brittle core assumption (monotonic progress). The method presumes $V(s_t, l)$ increases with $t$ inside successful demos. That is frequently false in multi-stage manipulation (staging, re-grasps, approach–retreat, exploration). The paper does not (i) justify the assumption’s scope, (ii) stress-test backtracking tasks, or (iii) validate $V$ against policy rollout success rather than within-demo ordering (which is tautological).

- Their potential shaping does not preserve optimal policy, opening the door to specification gaming. Potential-based shaping preserves optimal policies only when the potential is fixed, but the authors break the Ng–Harada–Russell invariance conditions by adding per-episode normalization and extra advantage terms. This creates room for specification gaming (reward hacking) because shaped returns need not preserve policy rankings (and hence optimal policy) w.r.t. task return. The paper neither analyzes these effects nor bounds failure modes. At minimum, the following would be required: (i) ablate normalization and the extra advantage terms, (ii) measure and report EPIC distance between the task reward and the implemented shaped reward. Omission of discussion on this matter is particularly problematic, since reward hacking is _the_ central issue when attempting to densify sparse rewards.

- Evidence is too weak to justify publication as an engineering result. Reported gains on CALVIN are tiny (≤1.63% relative; often <1%), with 3 seeds, and no significance testing, well within expected variance for these benchmarks. There is no real-robot result, and no head-to-head against VLM-reward or VLA-critic baselines. As an engineering paper, this fails the “convincing improvement” bar.

- Label-efficiency claims are overstated. While they avoid action labels, they still require successful demos and rely on a strong ordering prior; many prior methods also avoid action labels (e.g., T-REX/PTR) and are explicitly designed for non-monotone or suboptimal data. The manuscript likely oversells/mischaracterizes “no costly labels”, since they still need curated successful demos where the prior holds, which involves substantive supervision.

There are other minor issues (lack of detail in certain hyperparams for reproducibility, etc.), but they are minor in relation to the above.

**Questions:**

Please benchmark against similar work that densifies sparse VLA rewards? (most notably, VLAC; but other adjacent methods may be relevant, at the very least for discussion/related work: VLM-RM, RL-VLM-F, RoboCLIP, etc.).
Report EPIC distances between task reward and shaped reward; give any observed failure cases (loops/spec-gaming)?
Report results with >5 seeds with error bars?

---

### Official Review · Reviewer_4qSE · 2025-10-30

**Soundness:** 2
**Presentation:** 2
**Contribution:** 2
**Rating:** 2
**Confidence:** 4

**Summary:**

This paper proposes RFTF (Reinforcement Fine-Tuning with Temporal Feedback), a reinforcement learning framework for vision-language-action (VLA) models that aims to address the issue of sparse rewards in current reinforcement fine-tuning pipelines. The authors introduce a value model trained via contrastive learning on temporally ordered frames, assuming that state values should monotonically increase throughout a successful demonstration.

**Strengths:**

The paper identifies a legitimate limitation of existing reinforcement fine-tuning methods—sparse rewards—and attempts to address it through a self-supervised dense reward signal derived from temporal structure. The method does not require action labels, which could theoretically improve scalability.

**Weaknesses:**

* The “temporal monotonicity” assumption used to train the value model is overly simplistic and potentially invalid, as many tasks include reversible or non-monotonic progress (e.g., moving an object away before grasping).
* The contrastive objective directly mimics standard RLHF-style preference training without theoretical grounding.
* Moreover, the experimental gains are marginal and may fall within noise.
* There is no real-world validation and no comparison with other dense reward shaping approaches or learned reward models

**Questions:**

* How robust is the assumption that state values monotonically increase in expert trajectories? Have the authors quantified failure cases or counterexamples where this does not hold?
* How does the proposed contrastive value model compare to a simple temporal difference (TD) critic trained with pseudo-rewards or imitation rewards?
* What happens if the dense reward is noisy or inconsistent—does the PPO objective still converge?

---

### Official Review · Reviewer_y2XS · 2025-10-31

**Soundness:** 3
**Presentation:** 2
**Contribution:** 3
**Rating:** 6
**Confidence:** 3

**Summary:**

This paper discusses the challenge of sparse rewards in reinforcement fine-tuning for Vision-Language-Action (VLA) models, which limits their generalization and adaptation capabilities in embodied intelligence tasks. The paper proposes RFTF, a method that leverages a value model trained with temporal information to generate dense rewards without requiring robot action labels. The value model predicts state values based on human instructions and observations, and these values are integrated into a Proximal Policy Optimization (PPO) framework combined with reward shaping and Generalized Advantage Estimation (GAE). Experimental results on the CALVIN benchmark demonstrate state-of-the-art performance in the ABC-D generalization setting (average success length of 4.296) and rapid adaptation to unseen environments (average success length of 4.301 in environment D).

**Strengths:**

**Innovative Value Model Design**: The value model is trained using temporal information without robot action labels, reducing data dependency (Sec. 3.2). It achieves high accuracy (94%) after one epoch, ensuring reliable dense reward generation (Fig. 5). The architecture reuses VLA model weights for initialization, improving training efficiency (Sec. 3.2).

**Comprehensive Experimental Validation**: RFTF achieves state-of-the-art results on CALVIN ABC-D, outperforming baselines (Table 1). Adaptation experiments show significant performance gains in unseen environments (Table 2). Ablation studies demonstrate dense rewards' superiority over sparse rewards across generalization and adaptation settings (Table 3).

**Effective Integration of RL Techniques**: The combination of reward shaping (Eq. (2)) and GAE (Eq. (3)) addresses credit assignment in long episodes (Sec. 3.3). Techniques like KL divergence and clipping in the loss function (Eq. (4)) stabilize fine-tuning (Sec. 3.3). The method balances successful and failed samples via a coefficient η in the advantage function (Eq. (3)).

**Weaknesses:**

**Strong Assumptions in Value Model Training**: The value model assumes state values increase monotonically in expert demonstrations, which may not hold for complex, non-linear tasks (Sec. 3.2). No evidence is provided to validate this assumption across diverse tasks or failure cases (Fig. 2). The contrastive loss (Eq. (1)) relies solely on temporal ordering, ignoring task-specific reward structures.

**Limited Generalization**: Experiments are confined to the CALVIN simulation benchmark, with no real-world validation (Sec. 5). Comparison to other dense reward methods is absent, limiting contextualization (Sec. 4.2). Training details are omitted, hindering reproducibility (Sec. 4.1.4).

**Limited to Simulated Environment**: Validated only on the simulated CALVIN benchmark, with no real-world robot evaluation (Sec. 5). This limits assessment of practical applicability. The conclusion acknowledges this as a primary limitation, but no real-world validation makes it hard to gauge deployment readiness. Simulated environments differ significantly from real-world physics, perception, and dynamics.

**Questions:**

1.	How does the value model perform on tasks with non-monotonic progress, and could alternative training paradigms address this?
2.	What is the computational overhead of the value model during RL fine-tuning, and how does it scale with longer episodes?
3.	Are there specific environment attributes in CALVIN that made adaptation particularly effective, and would the method generalize to more dynamic environments?

---

### Note · Authors · 2025-11-28

I have read and agree with the venue's withdrawal policy on behalf of myself and my co-authors.